# Oocyst Shedding Dynamics in Children with Cryptosporidiosis: a Prospective Clinical Case Series in Ethiopia

Øystein H. Johansen,[a,b]* Alemseged Abdissa,[c,d] Ola Bjørang,[b] Mike Zangenberg,[e,f] Bizuwarek Sharew,[c] Yonas Alemu,[c] Sabrina Moyo,[a] Zeleke Mekonnen,[c] Nina Langeland,[a,g] Lucy J. Robertson,[h] Kurt Hanevik[a,g]

[a]Department of Clinical Science, University of Bergen, Bergen, Norway
[b]Department of Microbiology, Vestfold Hospital Trust, Tønsberg, Norway
[c]School of Medical Laboratory Sciences, Jimma University, Jimma, Ethiopia
[d]Armauer Hansen Research Institute, Addis Ababa, Ethiopia
[e]Department of Immunology and Microbiology, Centre for Medical Parasitology, University of Copenhagen, Copenhagen, Denmark
[f]Department of Infectious Diseases, Copenhagen University Hospital, Hvidovre, Denmark
[g]Norwegian National Advisory Unit on Tropical Infectious Diseases, Department of Medicine, Haukeland University Hospital, Bergen, Norway
[h]Parasitology, Department of Paraclinical Sciences, Faculty of Veterinary Medicine, Norwegian University of Life Sciences, Ås, Norway

**ABSTRACT** Knowledge on the duration of *Cryptosporidium* oocyst shedding, and how shedding may be affected by subtypes and clinical parameters, is limited. Reduced transmission may be a secondary benefit of cryptosporidiosis treatment in high-prevalence areas. We conducted a prospective clinical case series in children of <5 years presenting with diarrhea to a health center and a hospital in Ethiopia over an 18-month period. Stool samples were collected repeatedly from children diagnosed with cryptosporidiosis for up to 60 days. Samples were examined, and *Cryptosporidium* shedding was quantified, using auramine phenol, immunofluorescent antibody staining, and quantitative PCR (qPCR). In addition, species determination and subtyping were used to attempt to distinguish between new infections and ongoing shedding. Duration and quantity of shedding over time were estimated by time-to-event and quantitative models (sex- and age-adjusted). We also explored how diarrheal severity, acute malnutrition, and *Cryptosporidium* subtypes correlated with temporal shedding patterns. From 53 confirmed cryptosporidiosis cases, a median of 4 (range 1 to 5) follow-up stool samples were collected and tested for *Cryptosporidium*. The median duration of oocyst shedding was 31 days (95% confidence interval [CI], 26 to 36 days) after onset of diarrhea, with similar estimates from the quantitative models (31 days, 95% CI 27 to 37 days). Genotype shift occurred in 5 cases (9%). A 10-fold drop in quantity occurred per week for the first 4 weeks. Prolonged oocyst shedding is common in a pediatric clinical population with cryptosporidiosis. We suggest that future intervention trials should evaluate both clinical efficacy and total parasite shedding duration as trial endpoints.

**IMPORTANCE** Cryptosporidiosis is an important cause of diarrhea, malnutrition, and deaths in young children in low-income countries. The infection spreads from person to person. After infection, prolonged release of the *Cryptosporidium* parasite in stool (shedding) may contribute to further spread of the disease. If diagnosis and treatment are made available, diarrhea will be treated and deaths will be reduced. An added benefit may be to reduce transmission to others. However, shedding duration and its characteristics in children is not well known. We therefore investigated the duration of shedding in a group of young children who sought health care for diarrhea in a hospital and health center in Ethiopia. The study followed 53 children with cryptosporidiosis for 2 months. We found that, on average, children released the parasite for 31 days after the diarrhea episode started. Point-of-care treatment of cryptosporidiosis may therefore reduce onward spread of the *Cryptosporidium* parasite within communities and households.

Address correspondence to Øystein H. Johansen, haarklau@gmail.com.

*Present address: Øystein H. Johansen, Microbiology laboratory, Southern Health and Social Care Trust, Craigavon, Northern Ireland.

The authors declare no conflict of interest.

[This article was published on 14 June 2022 without accession numbers in the supplemental data set. The supplemental material was updated in the current version, posted on 13 July 2022.]

**KEYWORDS** *Cryptosporidium*, acute malnutrition, children, cryptosporidiosis, diarrhea, low-income setting, molecular subtyping, prolonged diarrhea, shedding, the CRYPTO-POC study

Cryptosporidiosis in sub-Saharan Africa mainly affects children under the age of 2 years and is largely caused by *Cryptosporidium hominis*. *Cryptosporidium parvum* is uncommon in livestock in Sub-Saharan Africa, and human *C. parvum* infections are caused almost exclusively by anthroponotic subtypes in this region (1). The strong association between health care-presenting cryptosporidiosis and risk of death in young children was firmly established by the Global Enteric Multicenter Study (GEMS) and the GEMS-1A follow-on study (2). The global health burden has been estimated to 13 million disability-adjusted life years lost every year, of which 67% are related to decreased growth in children who suffered from cryptosporidiosis in early childhood (3). As *Cryptosporidium* oocysts are robust, enabling persistence in the environment, and the infectious dose is low, transmission may be not only person-to-person or animal-to-person but also via transmission vehicles such as water, food, or fomites. *Cryptosporidium* infections that cause diarrhea are associated with higher quantities of oocysts being shed and pose an elevated risk of household transmission (4, 5). Secondary transmission rates may also be affected by the intensity and duration of shedding after an infection. The prolonged presence of *Cryptosporidium* in the stool may also mean that other potential enteropathogens in subsequent gut infections are overlooked.

Until a 2021 landmark publication from the Malnutrition and Enteric Disease Study (MAL-ED) (6), a few studies had indicated that oocyst shedding could continue for months after a cryptosporidiosis episode, in both adults and children, and a community cohort study in children growing up in a Peruvian shantytown (*n* = 25) reported a mean shedding duration of 17 days after onset of diarrhea (7–9). These studies were conducted in heterogenous populations and relied on low-sensitivity microscopy. The MAL-ED study, by contrast, investigated postdiarrheal shedding of a broad range of enteric pathogens, used sensitive real-time PCR detection, and was conducted in under-2-year-old children in diverse low-resource settings (6). The median duration of *Cryptosporidium* shedding was estimated to be more than 5 weeks after onset of the diarrheal episode. However, genotyping of follow-up samples was not done. Thus, later detections during the surveillance period may have represented new infections with other *Cryptosporidium* species or genotypes, with a risk of overestimating the median shedding duration. Furthermore, their estimate was based on analysis of cryptosporidiosis identified by community surveillance visits and is not necessarily valid for the population of children who are diagnosed in health care settings.

Targeted treatment for cryptosporidiosis is currently limited to nitazoxanide, a drug that is not effective in immunocompromised children and regrettably is FDA-approved only for children older than 12 months, although available data indicate no adverse effects in children aged 6 to 11 months (10–13). Significant progress has been made by several research groups in developing new pharmaceutical treatments (14), and an ongoing trial in Australian Aboriginal children may provide additional safety data for use of nitazoxanide in 3- to 5-month-olds (10). If prolonged duration of shedding is confirmed in the clinical setting, tangible secondary benefits of targeted cryptosporidiosis treatment may be to reduce transmission and to enable faster recovery of intestinal function (15, 16). We therefore need reliable estimates for the duration of oocyst shedding in those settings where interventions against cryptosporidiosis are most needed, i.e., low-resource health care settings. An additional point is that it should be possible to identify these shedders using simple and affordable, yet accurate, point-of-care tests.

Our study therefore aimed to describe the temporal dynamics of *Cryptosporidium* shedding, i.e., both overall duration of shedding and quantitative shedding patterns over time, supplemented by subtyping to help distinguish between new infections and ongoing shedding. Cases were identified by near-patient testing using light-emitting diode auramine-phenol fluorescence microscopy (LED-AP), already available in many low-income countries, as the method is used for tuberculosis point-of-care testing (17). Secondary objectives were to

explore how diarrheal severity, acute malnutrition, and *Cryptosporidium* species and subtypes correlated with shedding patterns over time.

We estimated the duration of shedding by two complementary approaches. First, we defined shedding as a binary event (ongoing shedding versus no longer shedding, as confirmed by microscopic detection and/or detection by quantitative PCR [qPCR]). Second, we considered shedding as a quantitative outcome, measured as DNA copies/g of stool in follow-up samples. In addition, cases with a shift in the detected *Cryptosporidium* genotype were classified as having "ongoing shedding" only up to the last positive detection before the genotype shift happened. To our knowledge, this is the first study of temporal oocyst shedding dynamics in human cryptosporidiosis to include subtyping.

## RESULTS

Cryptosporidiosis screening was performed by LED-AP microscopy on 878 diarrhea cases; 82 (9%) were positive, and, of these, 56 were eligible and consented to participate in the follow-up study. Shedding of *Cryptosporidium* DNA was confirmed in 54 of these 56 cases, as two cases had negative *Cryptosporidium* qPCR in both enrollment and follow-up stool samples (ID34 and ID40). One case that was shedding on enrollment failed to submit any follow-up samples (ID19) and was therefore excluded from the analysis. A median of 4 (range 1 to 5) follow-up stool samples were collected from each of the remaining participants. The lowest positive detection by the qPCR assay was 519 *Cryptosporidium* copies per gram of stool, corresponding to approximately 130 oocyst equivalents per gram, as there are 4 copies of the *cowp* gene per oocyst (i.e., 1 per sporozoite). See Table 1 for characteristics of the 53 participants included in the analysis of temporal shedding patterns and Fig. S1 for all microscopy, PCR, and genotyping findings for the 56 participants in the follow-up study.

Notably, two cases were still shedding oocysts at their last follow-up visits, 63 days after onset of the initial diarrheal episodes. Both were 6- to 11-month-olds, without acute malnutrition, who presented with acute diarrhea; HIV status was unknown for one and the other was HIV negative. A shift in *Cryptosporidium* genotype occurred in five cases (9%): in three cases, this was reflected by subsequent detections of a different *C. hominis gp60* allele family in samples taken 16, 24, and 25 days after onset of diarrhea, respectively (ID 9, 56, 28), in one case the shift happened after two consecutive negative samples, 34 days after onset of diarrhea (ID 48), and in one case, *C. hominis gp60* allele type Ib was detected in a sample obtained at day 37, after three consecutive samples with *C. parvum* IIc (ID 36). There were no cases where subtype shifted within the same *gp60* allele family.

**Duration of shedding.** As expected, the proportion of cases shedding *Cryptosporidium* declined over time. Figure 1 shows the time-to-event curves describing the decline in the proportion of cases shedding *Cryptosporidium* during the follow-up-period.

Stratification of the time-to-event curves by key subgroups revealed subtle differences by sex and age group, which might confound other comparisons. We therefore fitted a log-logistic parametric time-to-event model (Fig. S2) which allowed us to adjust for sex and age. The overall model of the decline in shedding is illustrated by Fig. 2, from which the estimated median duration of shedding was 31 days (95% confidence interval [CI], 26 to 36 days).

Shedding was protracted for several weeks in most cases; by 3 weeks after onset of diarrhea, only 21% (95% CI 11 to 32) had stopped shedding *Cryptosporidium*, rising to 75% (95% CI 62 to 87) by 6 weeks (Table 2).

Results from the parametric models, stratified by key predictors, are summarized in Table 3, where the odds ratios (OR) quantify differences in shedding duration across the follow-up period. There were no obvious differences by age group, sex, or acute malnutrition and only a trend toward shorter shedding duration for cases who were dehydrated on enrollment. No significant difference was found between *C. parvum* and *C. hominis*. However, when the shedding curves between the different *C. hominis gp60* subtype families were compared, *gp60* allele family Id, the second most common *gp60* allele family, was associated with a significantly longer duration of shedding than the other *C. hominis gp60* allele families.

**Change in *Cryptosporidium* shedding quantity over time.** Positive samples included in this analysis ranged in *Cryptosporidium* DNA quantity from 805 copies/g to 158 million

**TABLE 1** Characteristics of cryptosporidiosis cases in the follow-up study

| Characteristic | No. of cases (%)[a] |
|---|---|
| Age, in mo | |
| <6 | 1 (2) |
| 6–11 | 28 (53) |
| 12–23 | 22 (41) |
| 24–59 | 2 (4) |
| Sex | |
| Female | 25 (47) |
| Male | 28 (53) |
| Study site | |
| Jimma hospital | 41 (77) |
| Serbo health center | 12 (23) |
| Acute malnutrition | |
| Moderate acute malnutrition (MAM) | 4 (8) |
| Severe acute malnutrition (SAM) | 7 (13) |
| No acute malnutrition (NAM) | 42 (79) |
| HIV status[b] | |
| HIV positive | 0 (0) |
| HIV negative | 41 (100) |
| Diarrheal severity[c] | |
| No dehydration | 21 (43) |
| Some dehydration | 16 (33) |
| Severe dehydration | 12 (24) |
| Diarrheal duration (on enrollment) | |
| Acute (<7 days duration) | 37 (70) |
| Prolonged (7–13 days duration) | 13 (24) |
| Persistent (≥14 days duration) | 3 (6) |
| *Cryptosporidium* species and *gp60* allele family[d] | |
| *C. hominis* Ia | 14 (26) |
| *C. hominis* Id | 13 (25) |
| *C. hominis* Ib | 10 (19) |
| *C. hominis* Ie | 9 (17) |
| *C. parvum anthroponosum* IIc | 6 (11) |
| *C. parvum anthroponosum* IIc + *C. hominis* (nontypeable) | 1 (2) |
| *Cryptosporidium* spp. other than *C. parvum* or *C. hominis* | 0 (0) |

[a]*n* = 53, unless otherwise specified.
[b]*n* = 41; *n* = 12 missing HIV status.
[c]By DHAKA dehydration category; *n* = 49; *n* = 4 missing DHAKA score.
[d]Specific *gp60* subtype data can be found in Fig. S1.

copies/g. Quantities declined significantly with days since onset of diarrhea (smooth term, $P < 0.00001$) (Fig. 3).

The overall trend line was close to a quadratic relationship (effective degrees of freedom [EDF] = 1.9); however, for the first 4 weeks after onset of diarrhea, a striking pattern was seen where *Cryptosporidium* DNA shedding quantity dropped about 10-fold per week (Table S1). Thereafter, the overall trend line curved gently upwards, likely due to inclusion in the model of two cases who were still shedding after 63 days.

Although we observed some differences in shedding quantity between subgroups (Fig. 3), mirroring the subtle differences seen in the time-to-event graphs (Fig. 1), most subgroup differences were not significant at the 0.05 level, or significant only during a time window starting several weeks after onset of the diarrheal episode. The largest difference was seen at the subtype level, where *C. hominis gp60* allele family Id was associated with a higher quantity of shedding than the other *C. hominis* allele families during the time window from 11 to 55 days after onset of diarrhea (Table 3); see Fig. S3 for visualization of the models used to obtain these estimates.

We proceeded to estimate shedding duration by quantity over time by estimating the time point at which *Cryptosporidium* DNA quantity would drop below the qPCR detection

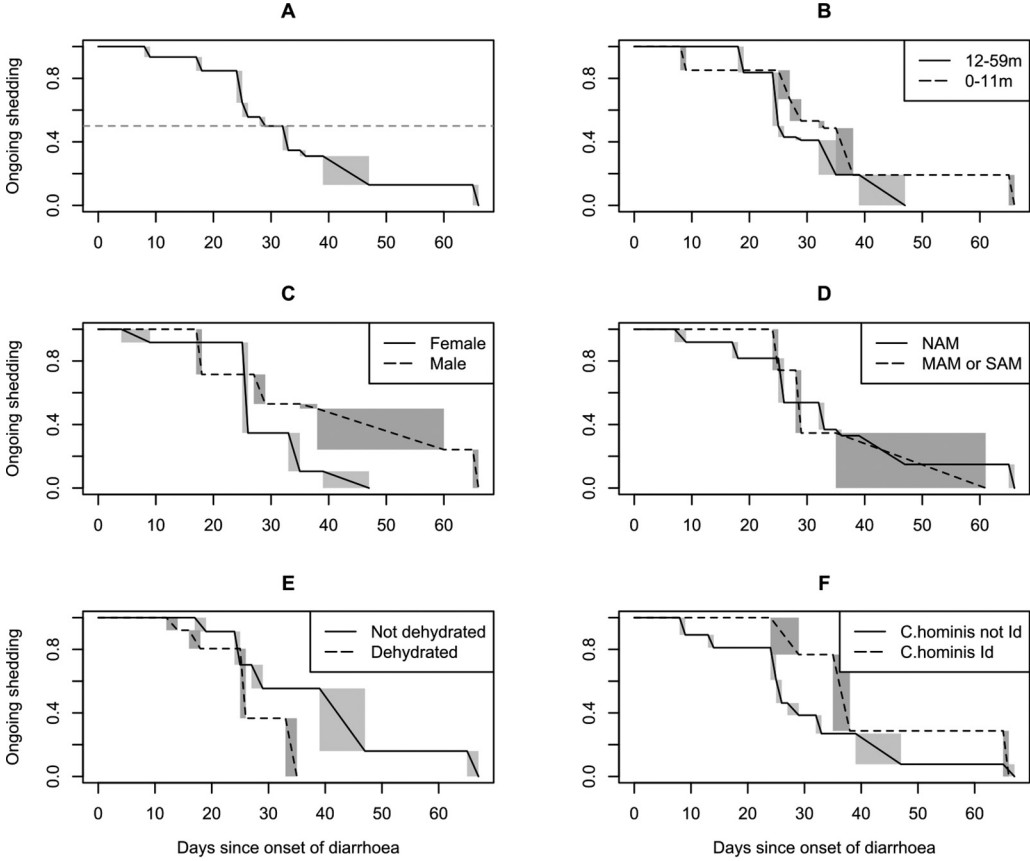

**FIG 1** Duration of *Cryptosporidium* shedding nonparametric time-to-event curves. Temporal decline in ongoing shedding; (A) overall, and stratified by (B) age, (C) sex, (D) acute malnutrition, (E) dehydration, and, (F) for *C. hominis* infections, *gp60* allele family. The shaded rectangles represent ranges of indeterminate shedding status, due to the interval-censored nature of the data. MAM/SAM/NAM, moderate/severe/no acute malnutrition.

threshold (Fig. S4). This yielded an estimate for the duration of *Cryptosporidium* DNA shedding at 31 days (95% CI from 25 to 37 days).

## DISCUSSION

*Cryptosporidium* parasite shedding was investigated by follow-up of individual children presenting to health care with cryptosporidiosis. Shedding duration was approximated using two complementary analytical approaches: first by time-to-event modeling where ongoing shedding was considered a binary outcome, based on microscopic detection of oocysts and/or DNA detection by PCR, and then by quantitative modeling of the drop in DNA quantity over time, using a highly sensitive qPCR assay. Both models yielded remarkably similar estimates for median duration of shedding, at 31 days, with 95% CIs from 26 to 36 days and 25 to 37 days, respectively. The findings demonstrate that prolonged shedding is common in a general pediatric clinical population with cryptosporidiosis. This is a slightly shorter estimate than, but still in keeping with, the 40-day estimate from the MAL-ED study, which was obtained using robust modeling and quantitative PCR for reliable case ascertainment (6). Although estimates obtained from a community study may not directly compare with our clinical population, it is worth pointing out that the MAL-ED study used a qPCR targeting the multicopy SSU rRNA gene, possibly with sensitivity better than that of our four-copy *cowp* qPCR assay at the low DNA quantities observed toward the tail end of an infection.

This is the first study of *Cryptosporidium* shedding in humans to include both species determination and subtyping. Genotype shift was seen in 5 of 53 cases. Previous estimates of shedding duration from studies where subtyping was not done may have been biased upwards. However, assuming that genotype shifts occur at a frequency similar to that which

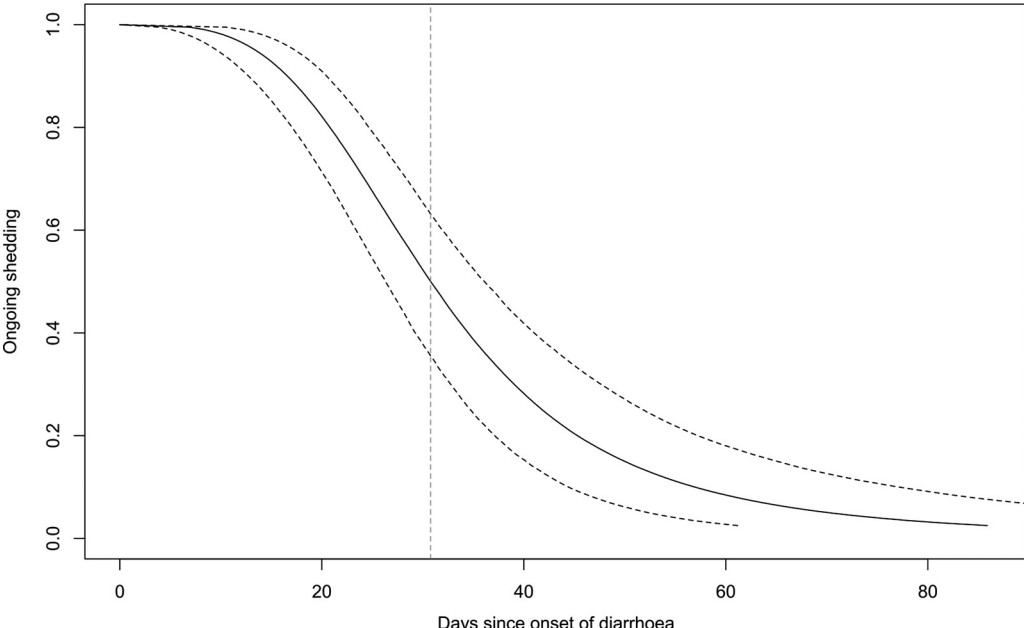

**FIG 2** Duration of *Cryptosporidium* shedding parametric time-to-event curve. Log-logistic time-to-event model, adjusted for sex and age. The dashed curves indicate 95% confidence intervals around the estimated proportion with ongoing shedding at a given time point; the vertical dashed line indicates median shedding duration.

we found, the effect of such bias is likely to be small. It is interesting that there were no *gp60* genotype shifts at the sub-allele-family level. As the *gp60* gene codes for highly immunoreactive surface molecules that are involved in adhesion and invasion of host cells (18), one possible explanation could be within-*gp60*-family acquired immunity.

Although differences in oocyst shedding quantities between *Cryptosporidium* species and genotypes have been reported previously (7, 19, 20), this has not been examined by quantitative modeling of the time course of an infection using repeated follow-up samples from individual children. Our findings indicate that quantitative trends can be subtle, unless considered across the timeline of the infection; for example, the association between *C. hominis* subtype Id and prolonged shedding was significant only during the middle and tail end of the infectious episode. It is far from clear whether this is caused by real parasite-level biological differences or whether the genotype is a marker of an unmeasured prognostic or confounding factor. Even if controlled experiments were to confirm differences in shedding density that manifest late in the course of infection, it is not clear how much this would contribute to overall transmission. Supported by the observed drop in quantity by about 10-fold per week for the first 4 weeks after onset of diarrhea, we would assume that the initial weeks of an infection contribute vastly more oocysts to the immediate environment. We can speculate that the early

**TABLE 2** Decline in shedding over time

| Days after onset of diarrhea | Proportion no longer shedding (%)[a] | 95% confidence interval (%) |
| --- | --- | --- |
| 7 | 1 | 0–2 |
| 14 | 6 | 2–12 |
| 21 | 21 | 11–32 |
| 28 | 42 | 29–56 |
| 35 | 61 | 48–76 |
| 42 | 75 | 62–87 |
| 49 | 84 | 72–93 |
| 56 | 89 | 79–96 |
| 63 | 93 | 84–98 |
| 70 | 95 | 88–99 |

[a]Estimated from the log-logistic time-to-event model, adjusted for age and sex.

**TABLE 3** Duration of *Cryptosporidium* shedding compared between groups

| Characteristic | Reference level | Odds ratio of no longer shedding, at any given time point (parametric model) | | | Test of difference (nonparametric model)[b] | Window of significantly different shedding quantity[c] |
| --- | --- | --- | --- | --- | --- | --- |
| | | | 95% CI for OR[a] | | | |
| | | OR | From | To | P value | |
| Age 0–11 mo | Age 12–59 mo | 1.07 | 0.76 | 1.50 | 0.32 | 22 to 52 days (higher) |
| Male | Female | 1.15 | 0.83 | 1.59 | 0.12 | 36 to 69 days (higher) |
| Acute malnutrition (MAM or SAM) | No acute malnutrition (NAM) | 0.97 | 0.68 | 1.38 | 0.70 | No window of significant difference |
| Any dehydration | No dehydration | 0.75 | 0.55 | 1.01 | 0.03 | 32 to 40 days (lower) |
| *C. parvum* | *C. hominis* | 0.94 | 0.57 | 1.55 | 0.31 | 43 to 69 days (higher) |
| *C. hominis gp60* allele family Id | *C. hominis gp60* allele family other than Id | 1.51 | 1.01 | 2.23 | 0.04 | 11 to 55 days (higher) |

[a]OR, odds ratios obtained from accelerated failure time log-logistic models, adjusted for sex and age (in months); age adjusted for sex only; sex adjusted for age only.
[b]Generalized log-rank test using within-subject resampling (not adjusted for sex and age).
[c]Window of significantly different shedding quantity (*Cryptosporidium* DNA copies/g, by qPCR) in days after onset of diarrhea; estimates obtained from generalized additive mixed model difference smooths, adjusted for sex and age (in months); age adjusted for sex only; sex adjusted for age only.

stage of infection is more likely to be associated with diarrhea with increased risk of onward transmission. If this is confirmed, health care-based interventions may be more effective than mass drug administration. To further investigate this question, we suggest that drug intervention studies should include repeat follow-up sampling with quantification and subtyping. Also, mathematical transmission modeling should account for the extended shedding duration of *Cryptosporidium* as well as symptoms and behavior.

Several limitations need to be kept in mind when interpreting our findings. First, our study size was limited to the number of positive cases in the CRYPTO-POC study that were within reach for follow-up. The overall sample size was decided for the primary objectives of the diagnostic accuracy study rather than for robust comparisons between groups; the observed subgroup differences have wide margins of uncertainty, should be considered trends at best, and need to be examined again in larger studies.

Second, although we attempted to account for confounding bias by adjusting the models for sex and age, this type of adjustment was not possible in the nonparametric time-to-event models. However, estimates of shedding duration from the nonparametric models were quite similar to those found in the adjusted parametric time-to-event models and to what was obtained by a generalized additive mixed model (GAMM), both of which included confounder adjustment.

Third, the observed tendency toward lower shedding in children who were dehydrated on enrollment seems counterintuitive but should at best be interpreted as a trend, as it was not a statistically significant finding in the adjusted models (Table 3).

Fourth, we interpreted repeat detection of the same *gp60* subtype as likely ongoing shedding from a single *Cryptosporidium* acquisition event. This is supported by the overall drop in quantity in repeat detections but is only a plausible assumption; another possibility is that some of the repeat detections represent new acquisitions of the same *gp60* genotype, e.g., from a still-present household source. Contact tracing has demonstrated clustering of subtypes in households (4, 5). However, determining the direction of transmission is likely to require a large-scale surveillance study with repeat stool sampling from all members of a household.

Last, although we found little evidence of mixed *gp60* sequences (data not shown), we may have missed some low-level mixed subtype infections. Elucidating this further would require the application of techniques to detect *gp60* subpopulations that are outside the scope of this analysis, e.g., deep sequencing, vector cloning, single-cell whole-genome amplification (21), or further validation of a promising new bioinformatics tool that can tease apart multistrain *C. parvum* infections by deconvoluting simple Sanger sequencing *gp60* chromatograms (22).

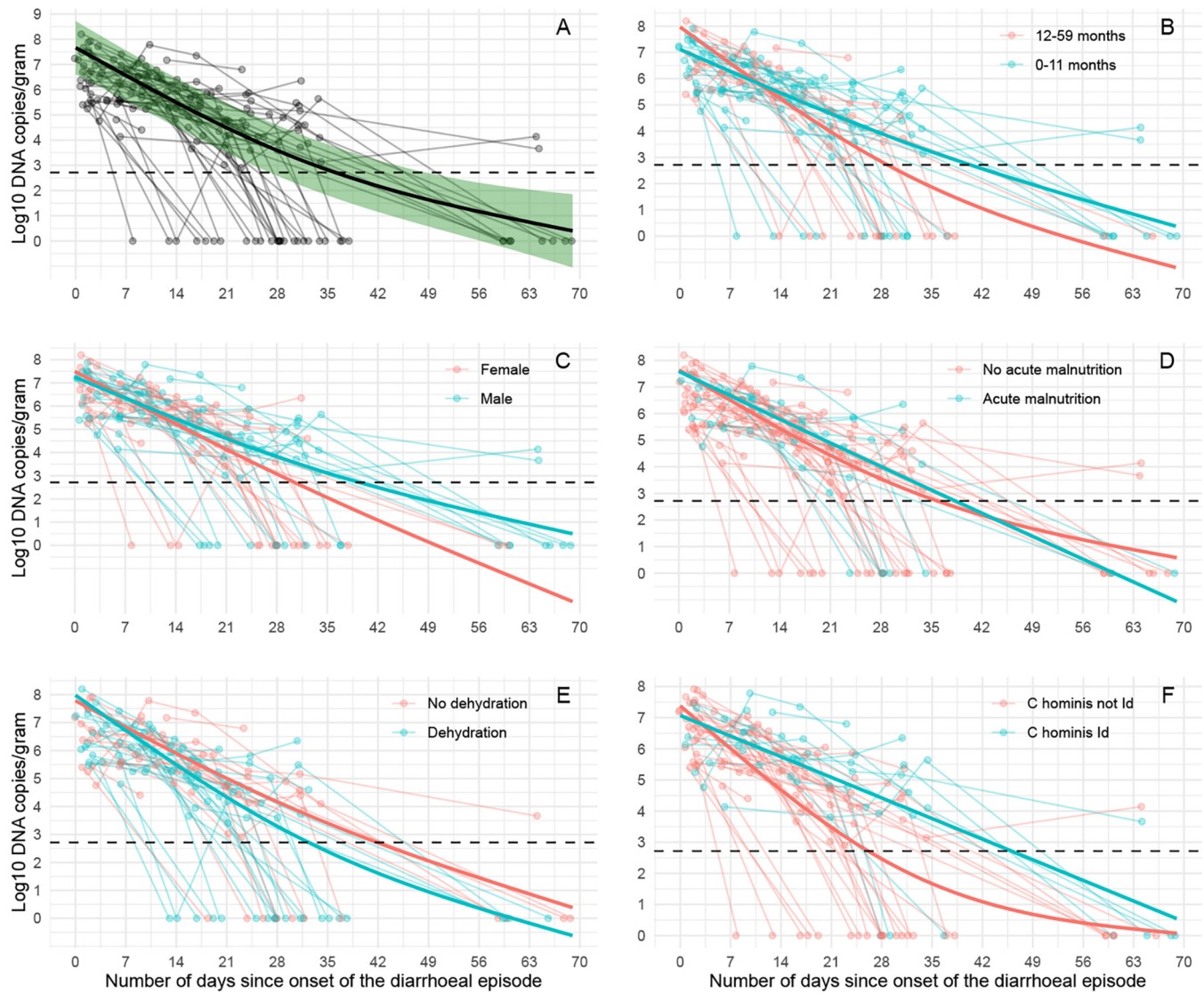

**FIG 3** Temporal patterns of *Cryptosporidium* shedding. *Cryptosporidium* DNA quantity in log$_{10}$ DNA copies/g; (A) overall, and stratified by (B) age, (C) sex, (D) acute malnutrition, (E) dehydration, and, (F) for *C. hominis* infections, *gp60* allele family. The dashed horizontal line represents the lowest reliable detection limit of the qPCR assay (519 copies/g). The shaded green area (A) represents the 95% confidence interval for the smoothed quantity estimate.

The main argument for offering testing and treatment for cryptosporidiosis in hospitals and health care centers is to reduce patient morbidity and mortality (23). In addition, targeted cryptosporidiosis treatment can be investigated as an intervention to reduce onward transmission (24) by controlling the key reservoir in children aged 6 to 24 months. Our cryptosporidiosis cases were diagnosed at the point of care, where a test-and-treat intervention might be practically feasible. Previous randomized controlled trials of nitazoxanide in children assessed parasitological clearance as a binary outcome by microscopy 7 to 10 days after initiating therapy, but total shedding duration was not a trial endpoint (25, 26). Ideally, future intervention trials of nitazoxanide or new therapeutics should include shedding duration as a separate endpoint in addition to clinical efficacy, supported by quantitative tests and subtyping. Based on the current and previous estimates of shedding duration (6), this will require minimum once-weekly follow-up of all cases for at least 6 weeks after onset of the diarrheal episode.

## MATERIALS AND METHODS

**Study design and participants.** The study was conducted from December 2016 to August 2018 in Jimma University Hospital (now Jimma Medical Centre) and Serbo Health Center in southwest Ethiopia as a follow-up study of a subset of participants in the CRYPTO-POC diagnostic accuracy study. Details of screening

and enrollment have been published previously (17). In brief, a child under 5 years of age was eligible for inclusion if they presented for health care, they had diarrhea (3 or more loose or looser-than-normal stools within the previous 24 h), and their caregivers consented to participation. Children who had been inpatients for longer than 24 h were already excluded from the CRYPTO-POC study to avoid bias of community prevalence estimates from nosocomial cases. In contrast with many other studies, children with less-severe diarrhea, prolonged diarrhea (≥7 days duration), or severe acute malnutrition (SAM) were included (17).

**Data collection.** Cryptosporidiosis point-of-care testing was performed by LED-AP microscopy (a semiquantitative test) at both sites, with results communicated back to the study nurse (17). Positive cases who resided within about 50 km of either site were invited to participate in a follow-up study that involved repeat clinical examination, interview, and stool samples. This started 7 days after enrollment, then continued weekly for the first 4 weeks, and ended with a final visit 60 days after enrollment. For *Cryptosporidium* oocysts, stool samples were examined by LED-AP and quantitative immunofluorescent stain antibody test (qIFAT) microscopy, and for *Cryptosporidium* DNA, stool samples were examined by a well-established quantitative PCR (qPCR) assay targeting the pan-species *Cryptosporidium cowp* gene. We generated standard curves using 10-fold dilutions of quantitative genomic *Cryptosporidium parvum* DNA, with final qPCR results expressed as DNA copies/g of wet stool, as described previously (17).

***Cryptosporidium* species determination.** On *Cryptosporidium* species qPCR-positive samples, *C. parvum* and *C. hominis* species identification was performed in a duplex real-time PCR assay targeting *C. parvum*- and *C. hominis*-specific sequences of the *lib13* gene, using primer and probe sequences from reference 27, but with probes adapted for use with the LightCycler system using locked nucleic acid probes (LC640) and a Förster resonance energy transfer quencher (BBQ-650) at the 3′ end. The PCR mixture contained LC FastStart DNA MasterPLUS HybProbe reaction mix (Roche), 625 nM each primer, 250 nM each probe, and 5 $\mu$L of DNA template, with water added to give a final reaction volume of 20 $\mu$L. This mixture was subjected to 50 cycles of denaturation at 95°C for 15 s, annealing at 60°C for 30 s, and extension at 72°C for 60 s, with an initial denaturation at 95°C for 10 min.

***Cryptosporidium* subtyping.** We sequenced the *Cryptosporidium gp60* gene, which codes for a surface glycoprotein that contains a conserved region that allows classification of *gp60* allele families and also a hypervariable serine repeat region that allows for classification into *gp60* allele family subtypes. The *gp60* PCR is a nested PCR, with primary and secondary primer sequences as described in reference 28. The reaction mixture for both rounds of PCR contained 2× Phire Hot Start II PCR master mix (ThermoScientific), 500 nM forward and reverse primer, and 5 $\mu$L of DNA template, with water added to give a final reaction volume of 20 $\mu$L. This mixture was subjected to 35 cycles of denaturation at 98°C for 5 s, annealing at 50°C for 5 s (primary PCR) and 55°C for 5 s (secondary PCR), and extension at 72°C for 15 s, with an initial denaturation at 98°C for 30 s and a final extension at 72°C for 1 min. Nested PCR products were purified using the Exo-CIP rapid PCR cleanup kit (New England BioLabs) according to the manufacturer's protocol. A total of 5 $\mu$L of purified template DNA and 5 $\mu$L of the forward or reverse primers used in the secondary PCR (at a concentration of 5 pmol/$\mu$L) were added to a LightRun 96-well plate for Sanger sequencing (Eurofins Genomics). Raw *gp60* gene sequence data were analyzed directly using the free software CryptoGenotyper (29). Sequences that were unsuccessfully typed or flagged for manual analysis by CryptoGenotyper were trimmed and aligned using Geneious Prime software (version 2021.1.1), where final sequences and phylogenetic trees were compared to reference sequences of valid *gp60* allele families, following the method outlined in reference 30, using the CryptoGenotyper curated list of *gp60* reference sequences (29) (accessed on 1 July 2021). This was supplemented by the U.S. National Centre for Biotechnology Information Basic Local Alignment Search Tool and subtype classification by manual inspection of the serine repeat region according to the nomenclature summarized in references 31 and 32.

**Dehydration score.** DHAKA (dehydration: assessing kids accurately) score was calculated based on assessment of general appearance, tears, skin pinch, and respirations; a score of 4 or more was deemed severe dehydration, 2 to 3 as some dehydration, and 0 to 1 as no dehydration (33).

**Acute malnutrition.** For children aged <6 months, severe acute malnutrition (SAM) was defined as weight-for-height z-score (WHZ) of ≤−3 of the WHO standard curves (34) and/or presence of bilateral edema involving at least the feet. Moderate acute malnutrition (MAM) was defined as a WHZ of ≤−2 and >−3 with no edema. Midupper arm circumference (MUAC) was used instead of WHZ for 6- to 59-month-olds, as it was difficult to bring height measurement boards to home visits (by motorcycle) and because MUAC is assumed to be less susceptible to dehydration than weight (35, 36); SAM was defined as MUAC of ≤115 mm and/or presence of bilateral edema involving at least the feet, and MAM was defined as MUAC of >115 mm and ≤125 mm with no edema.

***Cryptosporidium* shedding.** A diarrhea case was defined as shedding *Cryptosporidium* DNA during the enrollment episode if qPCR was positive in either the enrollment sample or the 1-week follow-up sample. A case was considered no longer shedding *Cryptosporidium* from the first day, after onset of the diarrheal episode, at which both *Cryptosporidium* qPCR and oocyst microscopy were negative. Genotype shift was defined as a detection of a new *Cryptosporidium* genotype (by species, *gp60* allele family, or *gp60* subtype) in a later sample. A case was considered having ongoing shedding at the latest time point, in days after onset of diarrhea, with either a positive *Cryptosporidium* qPCR or positive *Cryptosporidium* microscopy (LED-AP and/or qIFAT) but excluding those cases where a genotype shift had occurred.

**Time-to-event analysis.** Time to shedding cessation was the event of interest. The nonparametric maximum likelihood estimate for the time-to-event distribution was approximated by the Turnbull estimator, a generalization of the Kaplan-Meier estimate that accounts for interval-censored data, using the R package interval (37). We fitted various parametric accelerated failure time models to be able to get 95% confidence intervals for the median shedding duration, using the R package icenReg (38). The event-variable was time to shedding cessation, in days. The model was adjusted for sex and age (in months) by separate fixed effect terms. For comparison of time-to-event curves between subgroups, fixed effect regression coefficients were

exponentiated to obtain the odds ratio of difference in ongoing shedding across the follow-up period. We therefore also performed nonparametric tests for all comparisons using a generalized log-rank test, adapted to interval-censored data using within-subject resampling (39), using the R package interval (37).

**Modeling *Cryptosporidium* shedding quantity over time.** *Cryptosporidium* DNA quantity (measured in $\log_{10}$ copies/g of stool) in sequential follow-up samples was plotted against time (in days since onset of diarrhea) for each cryptosporidiosis case. The visual inspection indicated that the temporal drop in quantity followed a nonlinear pattern. We therefore fitted and visualized a trend line (i.e., an overall "best fit" aggregate curve representing the drop in quantity as a function of time) using a generalized additive mixed model (GAMM), using an identity link and thin plate regression splines, with the maximum number of base functions ("knots") set to 3, using the R package mgcv (40). The overall shape ("wiggliness") of the smoothed curve was quantified by the effective degrees of freedom (EDF) of the smooth term, where an EDF of 1 would imply a linear relationship, an EDF of 2 would imply a quadratic relationship, and an EDF of 3 would imply a cubic relationship, etc., and by significance of the smooth term by the *P* value. Trend line differences between key subgroups were visualized by adding to the GAMM grouping predictors for age group (0 to 11 months versus 12 to 59 months), sex, disease severity (dehydration, acute malnutrition), and *Cryptosporidium* genotype and by separate plotting of difference smooths with 95% confidence intervals, using the R package itsadug. Overall shedding duration was estimated from DNA quantity, where shedding cessation was defined as the time point at which *Cryptosporidium* DNA quantity dropped below the reliable detection limit of the qPCR. This was estimated by a separate GAMM where days since onset of diarrhea was the outcome variable and a smoothed trend line for DNA quantity was the main predictor. To all GAMMs were added random effect intercepts for each study participant to account for within-subject clustering and also separate fixed effect linear terms to adjust for confounding by sex and age.

**Ethics approval.** Jimma University IRB (reference: RPGC/610/2016), the Ethiopian National Research Ethics Review Committee (reference: JU JURPGD/839/2017), and the Regional Committee for Medical and Health Research Ethics of Western Norway (reference: 2016/1096) approved the study.

## SUPPLEMENTAL MATERIAL

Supplemental material is available online only.
**SUPPLEMENTAL FILE 1**, XLSX file, 0.02 MB.
**SUPPLEMENTAL FILE 2**, PDF file, 1 MB.

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
