## [Reviewer comments · Microbiology Spectrum]

Microbiology Spectrum

Oocyst shedding dynamics in children with cryptosporidiosis: a prospective clinical case series in Ethiopia

Oystein Johansen, Alemseged Abdissa, Ola Bjørang, Mike Zangenberg, Bizuwarek Sharew, Yonas Alemu, Sabrina Moyo, Zeleke Mekonnen, Nina Langeland, Lucy Robertson, and Kurt Hanevik

Corresponding Author(s): Oystein Johansen, University of Bergen

Review Timeline:

Submission Date:	January 6, 2022
Editorial Decision:	April 29, 2022
Revision Received:	May 24, 2022
Accepted:	May 25, 2022

Editor: Sumiti Vinayak

Reviewer(s): Disclosure of reviewer identity is with reference to reviewer comments included in decision letter(s). The following individuals involved in review of your submission have agreed to reveal their identity: David Carmena (Reviewer #1)

Transaction Report:

DOI: <https://doi.org/10.1128/spectrum.02741-21>

April 29, 2022

Dr. Oystein Haarklau Johansen
University of Bergen
Department of Clinical Science
Bergen
Norway

Re: Spectrum02741-21 (**Oocyst shedding dynamics in children with cryptosporidiosis: a prospective clinical case series in Ethiopia**)

Dear Dr. Oystein Haarklau Johansen:

Thank you for submitting your manuscript to Microbiology Spectrum. Your manuscript has been reviewed. Although the findings are interesting, there were some concerns raised that need to be addressed before a final decision can be made. In addition to addressing the reviewer comments, it will be useful to strengthen the introduction section. The significance and value of this study needs to be discussed by comparing what is known from other epidemiological studies such as GEMS-1A, Crypto-POC etc. Also, it will be useful for the reader if more information is provided in the introduction section on prevalence of *Cryptosporidium* species, *C. hominis* vs *C. parvum* (lines 60-62). Sensitivity of oocyst detection (detection limits) by qPCR also needs to be discussed in the results and discussion section. Moreover, the quality of the figures provided can be improved and high-resolution figures will be required.

Link Not Available

Sincerely,

Sumiti Vinayak

Journals Department
Reviewer comments:

Reviewer #1 (Comments for the Author):

The study by Johansen et al. aimed at investigating i) the temporal dynamics of the diarrhoea-causing enteric protozoan parasite

Cryptosporidium spp. in children under 5 years of age with diarrhoea in a low-income country (Ethiopia), ii) the potential correlation between oocyst shedding pattern over time and Cryptosporidium species/genotype and iii) the potential correlation between oocyst shedding pattern over time and pathogenicity measured as diarrhoea or malnutrition severity. Detection of the parasite was accomplished by auramine-phenol fluorescence microscopy. Samples that tested positive by this method were re-analysed by PCR and Sanger sequencing to determine Cryptosporidium species and subtypes. Oocyst shedding was measured by conventional (microscopy counting) and mathematical (quantitative model) methods. Duration of oocysts shedding was found to last around four weeks, with *C. hominis* gp60 allele family Id having statistically significant longer shedding patters during middle and late infection stages. Another interesting finding is the estimation of a 10-fold decrease in Cryptosporidium DNA shedding quantity per week of infection. Overall, this study is relevant because estimating the intensity and duration of Cryptosporidium oocyst shedding may influence secondary transmission rates, suggesting that early treatment of cases could be a practical approach to minimize transmission dynamics of the parasite in endemic areas. However, there are some issues that need addressing or clarification.

Major issues

1. One of the main conclusions of the study is the proposal that Cryptosporidium oocyst shedding should be monitored as secondary treatment outcome. What is, in the opinion of the Authors the practical feasibility of this intervention in the routine practice of low-resource clinical settings? This would involve contacting the infected children through time and collecting several stool specimens. I wonder if adherence to this sample collection protocol would be an issue.
2. Methods, line 131: the Authors selected a PCR method specifically designed for the specific detection of *C. hominis* and *C. parvum*. Whereas it is true that both specie account for approximately 90% of the human cases of cryptosporidiosis characterised globally, the remaining 10% are caused by less frequent Cryptosporidium species including *C. meleagridis*, *C. canis*, *C. felis*, and *C. ubiquitum*, among others. Considering that these species have been described circulating in different human African populations (see for instance Squire and Ryan, Parasit Vectors 2017;10:195; Muadica et al. 2021;10:255; or Messa et al. Pathogens 2021;10:452, among many others), please comment on how this species detection limitation could have affected to the results obtained in the present study.
3. Line 207: it is unclear (and this is probably due to my lack of knowledge on this point), how the Authors can reach an estimation of the DNA quantity using mathematical modelling without a sample or a series sample to be used as reference/pattern. I think it would be useful to explain this in a way understandable for a non-technical audience.

Minor issues

1. Abstract, line 39: please provide a median and a range of the number of successive stool samples collected and analysed per children.
2. Methods, line 113-114: definition of diarrhoea is already included in line 167. Please avoid duplicating information.
3. Line 142: gp60, as other gene abbreviations, should be italicised. Please amend here and through the whole manuscript (e.g. lines 143, 145, 156, 160, 161, 186, 247, 250, 252, 290, 291, 323, 350, 351, 353, and 402).
4. Lines 402-406: cloning in bacteria could be another way of detecting (and differentiating) mixed infections involving different Cryptosporidium species/genotypes. Please add.
5. Line 411: please note that nitazoxanide is only prescribed in children older than 12 months of age. I would recommend modifying to 12-24 months.

Staff Comments:

Preparing Revision Guidelines

Please return the manuscript within 60 days; if you cannot complete the modification within this time period, please contact me. If

you do not wish to modify the manuscript and prefer to submit it to another journal, please notify me of your decision immediately so that the manuscript may be formally withdrawn from consideration by Microbiology Spectrum.

POINT-BY-POINT RESPONSE TO REVIEWERS

(WITH DESCRIPTION OF CHANGES MADE IN THE MANUSCRIPT AND IN SUPPLEMENTARY MATERIAL)

Reviewer #1 (Comments for the Author):

Major issues

1.

One of the main conclusions of the study is the proposal that *Cryptosporidium* oocyst shedding should be monitored as secondary treatment outcome. What is, in the opinion of the Authors the practical feasibility of this intervention in the routine practice of low-resource clinical settings? This would involve contacting the infected children through time and collecting several stool specimens. I wonder if adherence to this sample collection protocol would be an issue.

ANSWER: Here we are specifically thinking about research trials of nitazoxanide or new pharmaceutical or nutritional interventions against cryptosporidiosis, rather than a suggestion for a change in current clinical practice. We see that our use of the term “secondary treatment outcome” is misleading, and have clarified this in both **Abstract:Conclusion** and in the last paragraph of the **Discussion**, where we also added some context about outcome measures that were used in previous RCTs for cryptosporidiosis.

2.

Methods, line 131: the Authors selected a PCR method specifically designed for the specific detection of *C. hominis* and *C. parvum*. Whereas it is true that both specie account for approximately 90% of the human cases of cryptosporidiosis characterised globally, the remaining 10% are caused by less frequent *Cryptosporidium* species including *C. meleagridis*, *C. canis*, *C. felis*, and *C. ubiquitum*, among others. Considering that these species have been described circulating in different human African populations (see for instance Squire and Ryan, *Parasit Vectors* 2017;10:195; Muadica et al. 2021;10:255; or Messa et al. *Pathogens* 2021;10:452, among many others), please comment on how this species detection limitation could have affected to the results obtained in the present study.

ANSWER: We appreciate this important point and fully agree that we should not assume that other species may not be important in our study area, and that they are therefore important to look for. For this reason, we did not limit the analysis to *C. parvum* and *C. hominis*, but we do appreciate that this is not communicated clearly enough in the manuscript.

The distinction between the primary non-species specific qPCR and the *C. parvum* and *C. hominis* specific (*lib13*) PCR has now been made clearer (in **Methods:Data collection**) by labelling the *Cryptosporidium* qPCR assay as detecting “*Cryptosporium* spp. DNA”, and by making it explicit that *lib13* PCR was a follow-up assay after the initial round of qPCR, and, also, by including a separate line in **Table 1** for “*Cryptosporidium* spp. other than *C. parvum* or *C. hominis*”. We hope these modifications make it clear for the reader that we looked for all species, but that we did not find any “non-parvum non-hominis” *Cryptosporidium* detections in this study.

(For information, had there been any C non-hominis non-parvum detections, we would have proceeded with *Cryptosporidium* 18s rDNA (*ssu*) sequencing in order to determine the species. However, this was not necessary for the purposes of the current study)

3.

Line 207: it is unclear (and this is probably due to my lack of knowledge on this point), how the Authors can reach an estimation of the DNA quantity using mathematical modelling without a sample or a series sample to be used as reference/pattern. I think it would be useful to explain this in a way understandable for a non-technical audience.

ANSWER: Good point, we have tried to clarify the key idea behind the method (in **Methods: Modelling Cryptosporidium shedding quantity over time**), which is basically to make a “best fit” aggregate curve based on the drop in quantity (on the y axis) as a function of time (on the x axis) , and by using language (e.g., “plotted”, “best fit”, “aggregate curve”) that we hope will prompt the reader to inspect the plots and appreciate the visual and fairly intuitive nature of the underlying statistical method.

We also added some detail on the qPCR assay to **Methods:Data collection**, including a specific mention that we used standard curves of serially diluted known quantities of *Cryptosporidium* DNA to calibrate the qPCR.

Minor issues

1.

Abstract, line 39: please provide a median and a range of the number of successive stool samples collected and analysed per children.

ANSWER: This has now been added to **Abstract:Results** and **Results:1st paragraph**.

2.

Methods, line 113-114: definition of diarrhoea is already included in line 167. Please avoid duplicating information.

ANSWER: The full definition is now provided at first mention in **Methods:Study design and participants**, and deleted from **Methods:Definitions**.

3.

Line 142: gp60, as other gene abbreviations, should be italicised. Please amend here and through the whole manuscript (e.g. lines 143, 145, 156, 160, 161, 186, 247, 250, 252, 290, 291, 323, 350, 351, 353, and 402).

ANSWER: Thank you for pointing this out, we have corrected all gene mentions to small letters and italics, including (i.e., *gp60* and *lib13*).

4.

Lines 402-406: cloning in bacteria could be another way of detecting (and differentiating) mixed infections involving different *Cryptosporidium* species/genotypes. Please add.

ANSWER: Good point, we have now mentioned both cloning in a vector, and a new and promising bioinformatics tool, that can be useful for elucidating mixed *gp60* infections, in the second last paragraph of **Discussion**.

5.

Line 411: please note that nitazoxanide is only prescribed in children older than 12 months of age. I would recommend modifying to 12-24 months.

ANSWER: Our remark was not intended to apply to just the use of nitazoxanide, but also to new therapeutics that may be developed in the future, but we appreciate that this was not clear. Furthermore, when it comes to therapeutic interventions, as our study is an observational study, we hope that it can inform further research (e.g., to provide useful data justifying the evaluation of shedding duration as a trial outcome) rather than a direct change in clinical practice. We have therefore rephrased the sentence (**Discussion:last paragraph**) to make this distinction clear to the reader.

On nitazoxanide specifically, the reviewer is here reminding us of the important fact that nitazoxanide is not currently FDA-approved for children 12 months or older. We think this is somewhat regrettable, as there is available clinical evidence of safety in trials that also included children younger than 12 months. Furthermore, the ongoing NICE-GUT trial in Australia is aiming to provide additional safety data in infants older than 3 months. We however acknowledge that this important context was missing from the manuscript, and it has now been provided in the **Introduction**.

RESPONSE TO EDITOR'S COMMENTS:

“In addition to addressing the reviewer comments, it will be useful to strengthen the introduction section. The significance and value of this study needs to be discussed by comparing what is known from other epidemiological studies such as GEMS-1A, Crypto-POC etc. Also, it will be useful for the reader if more information is provided in the introduction section on prevalence of Cryptosporidium species, *C. hominis* vs *C. parvum* (lines 60-62). Sensitivity of oocyst detection (detection limits) by qPCR also needs to be discussed in the results and discussion section. Moreover, the quality of the figures provided can be improved and high-resolution figures will be required.”

RESPONSE: The **Introduction** section has been expanded with some information on the molecular epidemiology of cryptosporidiosis in Africa, and some framing information referring to the invaluable information on risk association, morbidity, and mortality from the GEMS-1A and GBD studies (**Introduction:1st paragraph**).

We also provided more specific information on previous studies that have investigated shedding duration, to allow for comparison with the MAL-ED study on postdiarrhoeal shedding in a community setting, and to point out some features that distinguish these previous studies from the current study (**Introduction:2nd paragraph**).

Further details on the lowest reliable detection limit has now been added to **Results:1st paragraph** and we mention the possible impact of the different target in our qPCR assay compared with the target that was used in the MAL-ED study on postdiarrhoeal shedding (**Discussion:1st paragraph**)

Figure 1, 2 and 3 have now been removed from the main body of the text and is provided as separate files in high resolution tiff format. The original files, generated using the commonly used R ggplot2 package, are also available in scalable vector format (*.svg), which displays perfectly when imported into publishing software, e.g. Adobe photoshop, but the submission portal does not allow this format. Contact us if you want this instead of tiff format figures.

OTHER SIGNIFICANT CHANGES TO THE MANUSCRIPT:

Error correction 1: When revising the manuscript, we discovered that one of the participants in the follow-up study (ID 19), had in fact failed to submit any follow-up samples despite one documented follow-up visit. Although the impact on estimates would be small due to the interval-censoring methods uses, all statistical analyses were done again, after removing this participant from the analysis. Minor changes reflecting this drop from 54 to 53 participants, in the analysis of shedding duration, can now seen as minor corrections in **Results(1st paragraph), Table 1 , Table 2, Table 3, and in Supplementary dataset.**

Error correction 2:

While re-running all analyses, we discovered that one of the models, specifically the generalized additive mixed model difference smooths comparing males vs females (**Table 3, second row**), had not, in fact, been adjusted for age, as we intended. When this error was corrected, the estimated window of significant difference changed from 31-69 days to 36-69 days. This has now been corrected in **Table 3 and in Supplementary Figure S3.** All other models had been adjusted correctly, as stated in the manuscript text and Table footnotes.

We regret not having noticed these errors before the primary submission.

May 25, 2022

Dr. Oystein Haarklau Johansen
University of Bergen
Department of Clinical Science
Bergen
Norway

Re: Spectrum02741-21R1 (**Oocyst shedding dynamics in children with cryptosporidiosis: a prospective clinical case series in Ethiopia**)

Dear Dr. Oystein Haarklau Johansen:

I am pleased to inform you that your manuscript has been accepted, and I am forwarding it to the ASM Journals Department for publication. You will be notified when your proofs are ready to be viewed.

Sincerely,

Sumiti Vinayak
Editor, Microbiology Spectrum

Journals Department
Supplemental Material FOR publication: Accept
Supplemental Material FOR publication: Accept